# Pyrrolizidine Alkaloids Disturb Bile Acid Homeostasis in the Human Hepatoma Cell Line HepaRG

**DOI:** 10.3390/foods10010161

**Published:** 2021-01-14

**Authors:** Julia Waizenegger, Josephin Glück, Marcus Henricsson, Claudia Luckert, Albert Braeuning, Stefanie Hessel-Pras

**Affiliations:** 1Department of Food Safety, German Federal Institute for Risk Assessment, Max-Dohrn-Straße 8-10, 10589 Berlin, Germany; julia.waizenegger@web.de (J.W.); josephin.glueck@bfr.bund.de (J.G.); claudia.luckert@bfr.bund.de (C.L.); albert.braeuning@bfr.bund.de (A.B.); 2German Nutrition Society, Godesberger Allee 18, 53175 Bonn, Germany; 3Wallenberg Laboratory and Sahlgrenska Center for Cardiovascular and Metabolic Research, Institute of Medicine, University of Gothenburg, 413 45 Gothenburg, Sweden; marcus.henricsson@wlab.gu.se

**Keywords:** pyrrolizidine alkaloids, hepatotoxicity, HepaRG, cholestasis, bile acid

## Abstract

1,2-unsaturated pyrrolizidine alkaloids (PAs) belong to a group of secondary plant metabolites. Exposure to PA-contaminated feed and food may cause severe hepatotoxicity. A pathway possibly involved in PA toxicity is the disturbance of bile acid homeostasis. Therefore, in this study, the influence of four structurally different PAs on bile acid homeostasis was investigated after single (24 h) and repeated (14 days) exposure using the human hepatoma cell line HepaRG. PAs induce a downregulation of gene expression of various hepatobiliary transporters, enzymes involved in bile acid synthesis, and conjugation, as well as several transcription regulators in HepaRG cells. This repression may lead to a progressive impairment of bile acid homeostasis, having the potential to accumulate toxic bile acids. However, a significant intracellular and extracellular decrease in bile acids was determined, pointing to an overall inhibition of bile acid synthesis and transport. In summary, our data clearly show that PAs structure-dependently impair bile acid homeostasis and secretion by inhibiting the expression of relevant genes involved in bile acid homeostasis. Furthermore, important biliary efflux mechanisms seem to be disturbed due to PA exposure. These mole-cular mechanisms may play an important role in the development of severe liver damage in PA-intoxicated humans.

## 1. Introduction

Pyrrolizidine alkaloids comprise a large group of secondary plant compounds occurring ubiquitously in the plant kingdom. They are constitutively produced in about 3% of the world’s flowering plants as a protective mechanism against herbivores [1,2]. To date, over 660 structurally different pyrrolizidine alkaloids and their *N*-oxide derivates have been identified, and about half of them are considered to be toxic [3,4]. The PA-associated harmful effects on livestock and humans [5,6,7,8] are associated with a double bound in C1,2-position in the necine base. In the following, the abbreviation PA is always used for 1,2-unsaturated PAs. Intoxications in humans by PA-contaminated cereals, herbal teas, and herbal medicines were reported in the United States, India, Tajikistan, Afghanistan, and South Africa [9,10,11,12,13,14]. PAs can cause severe liver damage characterized by hemorrhagic liver necrosis, ascites, cirrhosis, and the development of the characteristic hepatic sinusoidal obstruction syndrome (HSOS) [6,14,15].

The molecular mechanisms of PA hepatotoxicity are not yet fully elucidated. A well-known molecular mechanism is the formation of adducts with DNA and proteins by reactive PA metabolites, which are formed by enzymatic conversion in the liver [7]. Further mechanisms at the molecular level, such as interactions of PAs with specific signaling and metabolic pathways, are not completely understood. Recently, a genome-wide expression study in primary human hepatocytes provided evidence for PA-mediated induction of apoptosis and impairment of bile acid homeostasis [16], suggesting a partial involvement of these mechanisms in PA-induced hepatotoxicity. Additionally, increased bile acid levels in blood, as well as disturbance of bile acid secretion, were shown in mice following an acute toxic dose of senecionine [17].

The bile produced in the liver is essential for the absorption and digestion of lipids from the intestinal lumen, as well as for the elimination of xenobiotics, endogenous compounds, and metabolic products (e.g., cholesterol, bilirubin, and hormones) [18,19]. It consists predominantly of water (82%), followed by bile acids, phospholipids, cholesterol, proteins, bilirubin, and electrolytes. Primary bile acids are synthesized from cholesterol in the hepatocytes by various enzymes (including cytochrome P450 monooxygenase 7A1 (CYP7A1), CYP8B1, and CYP27A1) [20,21]. Under physiological conditions, the primary bile acids are conjugated with the amino acids glycine or taurine, followed by secretion by hepatocytes into the bile canaliculi, along with the remaining bile constituents, and storage and concentration in the gallbladder. After release into the duodenum and subsequent absorption and digestion of the food constituents, about 95% of the bile acids are reabsorbed into the liver via blood circulation, thus entering enterohepatic circulation [20,22]. Therefore, hepatic transport proteins are essential for the formation of bile and maintaining bile flow. Sinusoidal (basolateral) transporters are responsible for the uptake of endogenous and exogenous substances into the hepatocytes, and canalicular (apical) transporters mediate the biliary secretion into the bile canaliculi [23,24]. If the secretion of bile acids and consequently the bile flow are impaired, potentially toxic bile components, such as bile acids, bilirubin, and cholesterol, can accumulate, and thus may lead to the damage of hepatic cells [25,26]. In 2013, Vinken et al. proposed an adverse outcome pathway (AOP) for drug-mediated cholestasis [26]. The proposed AOP connects as the primary molecular initiating event the inhibition of the canalicular transporter ATP-binding cassette subfamily B member 11 (ABCB11) with various key events (e.g., bile acid accumulation, regulation of nuclear receptors, induction of inflammation, and oxidative stress) and intermediate steps (e.g., induction of apoptosis/necrosis and gene expression changes) to the adverse outcome, cholestasis.

Based on the proposed AOP for cholestatic liver disease [26] and data suggesting that an impairment of bile acid homeostasis may contribute to the PA-induced hepatotoxicity [16,17], the present study aims to systematically investigate the effect of PA treatment on bile acid homeostasis in the human hepatoma cell line HepaRG. In this regard, possible structure-dependent effects were examined using the four structurally different PAs echimidine, heliotrine, senecionine, and senkirkine (for chemical structures see Figure 1). In addition, two different exposure scenarios were comparatively investigated, namely a single treatment for 24 h and a repeated treatment for 14 days. The human hepatoma cell line HepaRG was used due to its hepatocyte-like morphology, high metabolic activity, and its suitability to investigate bile acid homeostasis [21,27]. Furthermore, HepaRG cells can be cultivated under long-term conditions, enabling a continuous treatment with test substances [28,29].

## 2. Materials and Methods

### 2.1. Chemicals

Heliotrine was purchased from Latoxan SAS (Valence, France). Echimidine, senecionine, and senkirkine were purchased from PhytoLab GmbH & Co. KG (Vestenbergsgreuth, Germany). All other chemicals were purchased from Sigma-Aldrich (Taufkirchen, Germany) in the highest purity available.

### 2.2. Plasmids

The plasmid pGL4.14-CYP7A1-Prom was constructed as already described [30]. Briefly, the CYP7A1 promoter region was cloned into the vector pGL4.14 (Promega, Madison, WI, USA) upstream of the firefly luciferase reporter gene by means of sequence and ligation-independent cloning [31,32]. The CYP7A1 promoter (−2014 to −1 bp from translation start site) was amplified from human genomic DNA using the primers 5′-CGG TAC CTG AGC TCG CTA GCC AGG AAA GAA CTG CAC CCA TAA T-3′ and 5′-CAG ATC TTG ATA TCC TCG AGT TTG CAA ATC TAG GCC AAA ATC T-3′ and subsequently inserted between the NheI/XhoI site of pGL4.14. The construction of the Renilla luciferase expression plasmid pcDNA3-Rluc was described elsewhere [33].

### 2.3. Cell Culture

HepaRG cells were obtained from Biopredic International (Saint-Gregoire, France) and cultured in William’s E medium supplemented with 10% (*v/v*) fetal bovine serum (FBS) (both from Pan-Biotech GmbH, Aidenbach, Germany), 100 U/mL of penicillin, 100 µg/mL of streptomycin (Capricorn Scientific GmbH, Ebsdorfergrund, Germany), 5 µg/mL of insulin (Pan-Biotech GmbH, Aidenbach, Germany), and 5 × 10^−5^ M of hydrocortisone hemisuccinate (Sigma-Aldrich, Taufkirchen, Germany) at 37 °C in a humidified atmosphere containing 5% CO_2_. For all experiments, HepaRG cells were used in passages between 17 and 20. After seeding, HepaRG cells were cultivated in the medium for two weeks. To initiate differentiation of the cells, the HepaRG cells were then cultivated with a medium containing 1% dimethyl sulfoxide (DMSO) for two days, followed by a medium with 1.7% DMSO (Merck KGaA, Darmstadt, Germany) for a further 12 days. For investigating PA-induced hepatotoxic effects in HepaRG cells, two different incubation scenarios were performed: a single incubation for 24 h and a repeated incubation for 14 days comprising a total of seven incubations every two days. The exact cultivation and incubation schemes have recently been published [29].

The human hepatocarcinoma cell line HepG2 was obtained from the European Collection of Cell Cultures (ECACC, Porton Down, UK). The cells were grown in high-glucose Dulbecco’s modified Eagle’s medium (DMEM, PAN-Biotech GmbH, Aidenbach, Germany) supplemented with 10% (*v/v*) FBS (Capricorn Scientific GmbH, Ebsdorfergrund, Germany), 100 U/mL of penicillin, and 100 µg/mL of streptomycin (both from Capricorn Scientific GmbH, Ebsdorfergrund, Germany) at 37 °C in a humidified atmosphere containing 5% CO_2_. At a confluence of about 80–90%, the cells were passaged and plated at a density of 3 to 5 × 10^4^ cells/cm^2^. Cells at passages up to 12 were used for all experiments.

### 2.4. Preparation of RNA and Quantitative Real-Time PCR Analysis (qRT-PCR)

For gene expression analysis, HepaRG cells were seeded in six-well plates at a density of 2 × 10^5^ cells/well. After differentiation, cells were incubated with 5, 35, or 70 µM of PA or the solvent (1.7% DMSO, 0.7% acetonitril (ACN)) for either 24 h or 14 days. To investigate the effects of PAs on CYP7A1 gene expression in HepG2 cells, 7.5 × 10^5^ cells/well were seeded in six-well plates and incubated on the next day with the three abovementioned concentrations of PA for 24 h. Following incubation, HepaRG cells were harvested directly into an RLT buffer from the RNeasy Kit (Qiagen, Hilden, Germany) containing 1% β-mercaptoethanol, while HepG2 cells were harvested into phosphate-buffered saline (PBS) and centrifuged for 5 min at 300× *g*. After centrifugation, the supernatant was discarded, and the HepG2 cell pellet was dissolved in an RLT buffer containing 1% β-mercaptoethanol. The RNA of HepG2 and HepaRG cells was isolated using the RNeasy Mini Kit (Qiagen, Hilden, Germany). A total of 1 µg of RNA was reverse transcribed into single-stranded cDNA using the High Capacity cDNA Reverse Transcription Kit (Applied Biosystems, Foster City, CA, USA) according to the manufacturer’s protocol. Quantitative real-time PCR was performed on a 7900HT Fast Real-Time PCR System (Applied Biosystems, Darmstadt, Germany) using Maxima SYBR Green/ROX qPCR Master Mix (2×) (Thermo Fisher Scientific, Waltham, MA, USA). Thermal cycling conditions have been described elsewhere [16]. Relative quantification of gene expression was calculated according to the 2^−ΔΔCt^ method [34] by normalizing the Ct values of the PA-treated samples to that of the reference gene ACTB (encoding β-actin) and the solvent-treated samples. An upregulation of gene expression was represented by relative expression values > 1, while a downregulation was reflected by values in the range of 0 < x < 1. For allowing a dimension-matched expression for the up- and downregulation, the reciprocals of the values for downregulation were calculated and are therefore shown as values < −1.

### 2.5. Transcriptional Activation of CYP7A1 by Dual Luciferase Reporter Gene Assay

For the CYP7A1 reporter gene assay, HepG2 cells were seeded in 96-well plates at a density of 1.8 × 10^4^ cells/well. After 18–24 h, the cells were transiently transfected using the TransIT-LT1 transfection reagent (Mirus Bio, Madison, WI, USA) with 80 ng of the plasmid pGL4.14-CYP7A1-Prom and 1 ng of the Renilla luciferase expression plasmid (pcDNA3-Rluc) used as internal control for normalization. Four to 6 h after transfection, the cells were treated with four different concentrations of the PAs echimidine, heliotrine, senecionine, and senkirkine (5, 35, 70, or 250 µM) or the solvent (2.5% ACN) for 24 h. The known CYP7A1 promotor activity inhibitor phorbol 12-myristate 13-acetate (PMA, 5 µM) was used as a positive control [35]. After 24 h, the cells were lysed by adding 50 µL of lysis buffer to each well (100 mM of potassium phosphate with 0.2% (*v/v*) Triton X-100, pH = 7) and incubated for 15 min on a plate shaker. After centrifugation, luciferase activity was analyzed as previously described [36]. Firefly luciferase values were normalized to Renilla luciferase values and expressed as fold-induction referred to solvent control.

### 2.6. Staining of Bile Canaliculi to Assay Canalicular Efflux

The fluorescent dye CDFDA was used to investigate canalicular efflux in HepaRG cells. The membrane-permeable CDFDA is metabolized by intracellular esterases to CDF, a substrate of ABCC2, resulting in fluorescence labeling of bile canaliculi. Therefore, 5.5 × 10^4^ HepaRG cells/well were seeded in 24-well plates. After differentiation, the cells were treated with a PA (5 or 35 µM), the solvent (1.7% DMSO, 0.35% ACN), or the positive control (10 mM of acetaminophen (APAP)) for 24 h or 14 days. Cells were washed three times with Hank’s balanced salt solution buffer (HBSS; 5.4 mM of KCl, 0.44 mM of KH_2_PO_4_, 0.5 mM of MgCl_2_ × 6 H_2_O, 0.41 mM of MgSO_4_ × 7 H_2_O, 137 mM of NaCl, 4.2 mM of NaHCO_3_, 0.34 mM of Na_2_HPO_4_, 25 mM of D-glucose, and 10 mM of 4-(2-hydroxyethyl)-1-piperazineethanesulfonic acid (HEPES, pH = 7.4)) prior to incubation with 5 µM of CDFDA for 30 min at 37 °C. Supernatants were discarded and cells were washed again three times with an HBSS buffer. The distribution of CDF was analyzed using the fluorescence microscope Axio Observer.D1 (objective EC Plan-Neofluar 5x/0.16 Ph 1) at λ_ex/em_ = 470/525 nm.

### 2.7. Staining of the Tight Junction Protein Zonula Occludens-1 (ZO-1) Combined with Nuclei Staining

HepaRG cells were seeded in 24-well plates on gelatin-coated cover slips (Ø 13 mm) at a density of 5.5 × 10^4^ cells/well to determine effects of PAs on tight junction proteins. Subsequent to the treatment with either 5 or 35 µM of PA (echimidine, heliotrine, senecionine, and senkirkine), the solvent (1.7% DMSO, 0.35% ACN), or the positive control APAP (10 mM) for 24 h, cells were washed twice with PBS and fixed by treatment with ice-cold methanol for 20 min. After two washing steps with PBS, nuclei were stained by using SYTOX™ Orange Nucleic Acid Stain (1 µM) (Thermo Fisher Scientific, Braunschweig, Germany), and cells were washed again twice with PBS. Subsequently, cells were blocked with 10% bovine serum albumin (BSA) in PBS for 1 h at room temperature. The cells were washed twice with PBS before incubation with the primary antibody ZO-1 (1:400 in blocking solution) (Cell Signaling Technology Europe, B.V., Leiden, Netherlands) for 4 h at room temperature. Following primary antibody incubation, the cells were washed twice in PBS and afterwards incubated with the secondary antibody AlexaFlour488-conjugated Goat Anti-Rabbit IgG Alexa Fluor 488 secondary antibody (1:400 in blocking solution; Thermo Fisher Scientific, Waltham, MA, USA) for 1 h in the dark at room temperature. Coverslips were washed twice in PBS and once in H_2_O and then mounted onto slides using Vecta Shield HardSet Mounting Medium (Vector Laboratories, Burlingame, CA, USA). Imaging of SYTOX Orange (λ_ex/em_ = 547/570) and ZO-1/Alexa Fluor 488 (λ_ex/em_ = 488/519) was performed using the confocal laser scanning microscope Leica TCS SP5 (objective HCX PL APO 63x/1.40 OIL PH3 CS).

### 2.8. Analysis of Bile Acid Content

The amount of different bile acids was detected via ultra-performance liquid chromatography-tandem mass spectrometry (UPLC-MS/MS) in cell culture supernatants and cell lysates. Therefore, HepaRG cells were seeded at a density of 0.2 × 10^5^ cells/well in six-well plates. After the proliferation and differentiation period of four weeks, the FBS level in the medium was reduced to 2% for 48 h. The cells were then washed once with pre-warmed PBS and incubated with either 5, 21, or 35 µM of PA, the solvent (1.7% DMSO, 0.35% ACN), or the positive control (20 µM of cyclosporine A) in an FBS-free medium to avoid interactions with bovine bile acids occurring in FBS [21]. The amount of cell culture medium was reduced to 1 mL per well to yield a higher concentration of secreted bile acids. Supernatants were collected after incubation for 48 h. Cells were washed twice with PBS, trypsinized for 15 min at 37 °C, and collected by adding 1 mL of PBS. After centrifugation at 250× *g* for 5 min, the supernatant was discarded. To increase the amount of bile acids, the cells and medium of three wells were pooled.

Bile acids were quantified using UPLC-MS/MS as described previously [37]. Briefly, bile acids in cell lysates and cell culture supernatants were extracted by adding methanol containing deuterated internal standards. After intensive mixing and centrifugation, methanol was evaporated under a stream of nitrogen. The samples were resuspended in a 1:1 (*v/v*) methanol:water mixture and injected onto the UPLC system (Infinity1290, Agilent Technologies, Santa Clara, CA, USA). Separation was performed on a Kinetex C18 column (Phenomenex, Torrance, CA, USA) using water and acetonitrile as mobile phases. Detection was operated in negative mode using a QTRAP 5500 mass spectrometer (Sciex, Concord, ON, Canada).

### 2.9. Statistical Analysis

Statistical analysis was performed with SigmaPlot 13.0 software. Statistical differences between the mean values of the treatments and the solvent control were determined by one-way ANOVA, followed by Dunnett’s post hoc test. Statistically significant differences were assumed at *p* < 0.05.

## 3. Results

### 3.1. PA-Dependent Alterations of Gene Expression of Transporters, Enzymes, and Transcription Regulators Involved in Bile Acid Homeostasis

Considering the microarray data of Luckert et al. (2015) [16] and the proposed mechanisms for the development of cholestasis according to the AOP for cholestatic liver diseases [26], 32 target genes were selected to examine possible effects on their expression after 24 h and 14 days of treatment with the four structurally different PAs echimidine, heliotrine, senecionine, and senkirkine via qRT-PCR. These 32 target genes comprise 14 hepatobiliary transporters, eight enzymes, and ten transcriptional regulators involved in bile acid homeostasis. Data of cell viability were recently summarized in Waizenegger et al. (2018) [29]. Based on these results, the following three concentrations were chosen: 5 μM (non-cytotoxic), 35 μM (non- to slightly cytotoxic), and 70 μM (cytotoxic). The qRT-PCR results showed that only the higher concentrations (35 and 70 μM) affected gene expression, whereas no significant changes occurred after treatment with the lowest concentration (5 μM). Furthermore, the regulatory effects after 24 h of PA treatment seemed to be higher than after 14 days. Additionally, the strongest regulatory effects were found after treatment with the retronecine-type PA echimidine and senecionine, while the weakest were detected for the heliotridine-type PA heliotrine.

The gene expression data showed a significant downregulation of five ABC transporters (*ABCB4*, *ABCB11*, *ABCC2*, *ABCC3*, *ABCC6*) and six solute carrier (SLC) transporters (*SLC10A1*, *SLC22A7*, *SLC22A9*, *SLC51A*, *SLCO1B1*, and *SLCO2B1*). After 24 h of PA treatment (Figure 2A), the most pronounced effects were found for the three SLC transporters *SLC22A7*, *SLC22A9*, and *SLC51A*, followed by the two SLC transporters *SLC10A1* and *SLCO2B1*. A complete list of the gene expression values is provided in Appendix A. A lower but significantly reduced gene expression was observed for the three ABC transporters *ABCB4*, *ABCB11*, and *ABCC6*. The weakest significant regulatory effects were detected for the two ABC transporters *ABCC2* and *ABCC3*. For the two SLC transporters *SLC51B* and *SLCO1B3*, no significant changes in gene expression were observed, except for the treatment with 35 and 70 μM of senecionine, as well as 70 μM of heliotrine. In comparison, after 14 days of treatment, the strongest decrease in gene expression was also observed for the SLC transporters *SLC22A7* and *SLC51A*, while the regulatory effect on *SLC22A9*, as well as *SLCO1B1* and *SLCO2B1*, was significantly lower compared to the 24 h treatment (Figure 2A). In contrast, *ABCB11* was more downregulated after 14 days of PA treatment than after 24 h. The weakest significant downregulation was detected for the ABC transporters *ABCB4*, *ABCC2*, and *ABCC6* and the SLC transporter *SLC10A1*, whereas no significant effects were found for the three transporters *ABCC3*, *SLC51B*, and *SLCO1B3* after continuous PA treatment. Finally, for the transporter *ABCB1*, a significant downregulation of gene expression was only observed after 14 days of treatment with the retronecine-type PAs echimidine and senecionine.

Concerning the effects of PAs on several enzymes involved in bile acid homeostasis, gene expression data revealed the strongest downregulation for the three CYP monooxygenases *CYP3A4*, *CYP7A1*, and *CYP8B1* after 24 h of PA treatment, with the most prominent repression for *CYP7A1*, the rate-limiting enzyme in bile acid formation (see Figure 2B). Furthermore, a significant downregulation was found for the phase II enzymes *SULT2A1* (encoding sulfotransferase 2A1) and *UGT2B4* (encoding UDP-glucuronosyltransferase 2B4), as well as for *BAAT* (encoding bile acid-CoA:amino acid *N*-acyltransferase). All three enzymes are required for bile acid conjugation. Compared to the aforementioned enzymes, gene expression of the two CYP monooxygenases *CYP27A1* and *CYP39A1*, also involved in bile acid synthesis, was only slightly decreased after PA treatment. In line with the 24 h treatment, the 14-day treatment also led to the strongest decrease in gene expression of *CYP7A1*, followed by *CYP3A4* and *CYP8B1*. However, the downregulation of *BAAT*, *SULT2A1*, and *UGT2B4* was less pronounced after 14 days. In contrast to the 24 h PA treatment, no effect was found on the expression of *CYP27A1* and *CYP39A1* after 14 days.

Additionally, for both PA treatment schemes, gene expression data showed a downregulation of all 10 investigated transcription regulators (see Figure 2C). After 24 h, the strongest downregulation was observed for *NR1I3* (encoding the constitutive androstane receptor), followed by *NR1I2* (encoding the pregnane X receptor) and further on *HNF4A* (encoding the hepatocyte nuclear factor 4 alpha), *PPARA* (encoding the peroxisome proliferator activated receptor alpha), and *NR0B2* (encoding small heterodimer partner). The weakest downregulation was found for the five remaining transcription regulators *ESR1* (encoding the estrogen receptor alpha), *NR1H4* (encoding the farnesoid X receptor), *HNF1A* (encoding the hepatocyte nuclear factor 1 alpha), *INSIG2* (encoding the insulin induced gene 2), and *SREBF1* (encoding the sterol regulatory element-binding transcription factor 1). After 14 days of PA treatment, the strongest downregulation was also detected for *NR1I3*, followed by *NR0B2*, *NR1I2*, and *SREBF1*. In line with the 24 h treatment, *NR1H4*, *HNF4A*, and *INSIG2* were only slightly downregulated after 14 days of PA treatment, whereas no regulation of gene expression was observed for *ESR1*, *HNF1A*, and *PPARA*.

### 3.2. PA-Dependent Inhibition of CYP7A1 Promoter Activity and Gene Expression

Since PA treatment in HepaRG cells led to a strong decrease in *CYP7A1* gene expression (see Section 3.1), possible inhibitory effects of PAs on the transcriptional activity of *CYP7A1* were investigated using a reporter gene assay. Therefore, HepG2 cells were transfected with the reporter gene plasmid pGL4.14-CYP7A1-Prom and the control plasmid pcDNA3-RLuc. Following 24 h of PA treatment (5, 35, 70, or 250 μM), the firefly and Renilla luciferase activities were determined. In HepG2 cells, even the highest PA concentration did not induce cytotoxicity. This might be due to the low expression level of phase I xenobiotic-metabolizing enzymes. As shown in Figure 3, no concentration-dependent inhibition of *CYP7A1* promoter activity was observed for the four PAs. Although there was a significant decrease in firefly luciferase activity after exposure to the PAs echimidine and heliotrine, this inhibition was relatively constant across all four investigated concentrations (~1.5- and 1.9-fold for 5 µM and 250 µM of echimidine, respectively, and ~1.6- and 1.8-fold for 5 µM and 250 µM of heliotrine, respectively).

The treatment with the known inhibitor of *CYP7A1* promoter activity PMA resulted in a significant inhibition of transcriptional activity by about 2.5-fold. Furthermore, this substance showed a concentration-dependent repression. To investigate whether the effect on *CYP7A1* repression by PAs is dependent on the metabolism of the PA, the influence of the four PAs on *CYP7A1* gene expression in HepG2 cells was investigated. Treatment for 24 h with the retronecine-type PAs echimidine and senecionine did not lead to a significant change in *CYP7A1* gene expression in HepG2 cells, whereas treatment with 70 µM of the PAs heliotrine and senkirkine resulted in a weak but significant downregulation of 1.9- or 1.5-fold, respectively (data depicted in Appendix A).

### 3.3. Effect of PAs on ABCC2-Driven Canalicular Efflux

In order to examine whether PA treatment for 24 h or 14 days affects bile flow, HepaRG cells were incubated with CDFDA and analyzed by fluorescence microscopy. Since CDFDA is intracellularly converted by esterases to the fluorescent ABCC2 substrate CDF, ABCC2-mediated export of CDF into the bile canaliculi leads to their fluorescence labeling [38]. As shown in Figure 4 and Appendix A, after treatment with the solvent control and the low concentration (5 μM) of the retronecine-type PAs echimidine and senecionine, CDF fluorescence was visible as clear, distinct green dots revealing an intact ABCC2 transport system. In contrast, treatment with the higher concentration of echimidine and senecionine (35 μM) for 24 h and 14 days led to a substantial effect on the transport and distribution of CDF (Figure 4B). The staining of bile canaliculi appeared rather diffuse and indistinct, with more extension across the area, indicating an accumulation of the dye in the bile canaliculi. After treatment with the heliotridine-type PA heliotrine and the otonecine-type PA senkirkine for 24 h, no effects on CDF transport were observed (Appendix A), and the staining of the bile canaliculi was comparable to the solvent control. However, after 14 days of treatment, heliotrine (35 μM) showed a slight impairment of CDF transport, whereas senkirkine had no effect after prolonged exposure (Appendix A). In summary, in particular, the retronecine-type PA seem to affect the transport of CDF and thus possibly the bile acid flow.

### 3.4. Influence of PAs on the Tight Junction Protein ZO-1

According to the proposed AOP by Vinken et al. (2013), the impairment of tight junctions is one of the secondary molecular events that may contribute to the development of cholestatic liver disease [26]. The localization of the tight junction protein ZO-1 was investigated in more detail using immunofluorescence microscopy to check a possible influence of PA treatment on cell–cell contacts. As depicted in Figure 5, only the high concentration of the retronecine-type PA senecionine (35 μM) showed slight changes in immunofluorescence of the ZO-1 protein after 24 h compared to the solvent control, highlighted by discontinuities in the grid structure and a decrease in the intensity of the green immunofluorescence similar to the positive control (10 mM of APAP). No influence on the tight junction protein ZO-1 was observed for the treatments with lower concentration (5 µM) or for the treatment with 35 μM of the other three PAs echimidine, heliotrine, and senkirkine.

### 3.5. Effects of PAs on Bile Acid Content

Amounts of primary and conjugated bile acids were measured in HepaRG cells and supernatants after treatment for 48 h with 5, 21, or 35 µM of PA. The primary bile acids synthesized in the liver are cholic and chenodeoxycholic acid, which are further conjugated with glycine or taurine. Within the UPLC/MS method used, the following bile acids were measured: primary bile acids, such as cholic acid and chenodeoxycholic acid, as well as conjugated bile acids, such as taurohyocholic acid, taurocholic acid, taurochenodeoxycholic acid, taurodeoxycholic acid, glycohyocholic acid, glycocholic acid, glycochenodeoxycholic acid, and glycodeoxycholic acid.

For a better comparison of the intracellular and extracellular effects, total bile acid amounts were referred to the solvent control (0.35% ACN), which was set as 100%. The sum of the determined bile acids in each treatment group is depicted in Figure 6. An overall concentration-dependent decrease of bile acids was observed intra- and extracellularly after PA incubation. However, the strongest decrease occurred for the retronecine-type PAs senecionine and echimidine. The treatment of HepaRG cells with the known cholestasis inducer cyclosporine A also resulted in a decrease to 50% and 20% of the extra- and intracellular bile acid content, respectively.

## 4. Discussion

A balanced bile acid homeostasis is a basic requirement for healthy normal liver function, including enzymes involved in the production and detoxification of bile acids, as well as sinusoidal and canalicular transporters mediating the bile acid flow. Recent research results point to an involvement of impaired bile acid homeostasis in PA-induced hepatotoxicity [16,17]. Therefore, the present study initially investigated the influence of PA treatment on the gene expression of 14 hepatobiliary transporters by qRT-PCR. This revealed a significant downregulation of gene expression of five ABC transporters and six sinusoidal SLC transporters. Additionally, the expression of several enzymes involved in bile acid homeostasis, synthesis, and detoxification was analyzed, also showing a significant PA-dependent downregulation. Furthermore, the effects of PA treatment on the gene expression of various transcriptional regulators were determined. A clear repression of the transcription regulators was shown, especially for *NR1I3* and *NR1I2*, followed by *HNF4A* and *NR0B2*. In general, these effects were more pronounced after 24 h of PA treatment than after 14 days of PA treatment. In regard to the chemical structure of the PA, the strongest effects were observed for the retronecine-type PAs echimidine and senecionine, whereas heliotridine-type PA heliotrine exerted the weakest effects. Similar results regarding the influence on gene expression of hepatobiliary transporters and also of enzymes involved in bile acid metabolism could be observed by Luckert et al. (2015) in primary human hepatocytes [16]. After treatment with 100 µM of the PAs echimidine, heliotrine, senecionine, and senkirkine, a downregulation of the gene expression of *SLC10A1*, *SLC22A7*, *SLCO1B1*, and *SLCO2B1* was observed. In accordance with the results presented here, Luckert et al. (2015) also observed a PA-related downregulation of the canalicular transporters *ABCC2* and *ABCB11* in primary human hepatocytes in their microarray analysis. The authors also observed a PA-induced downregulation for the enzymes *BAAT*, *CYP7A1*, *CYP8B1*, *CYP27A1*, and *SULT2A4* in primary human hepatocytes. Furthermore, the bioinformatic analysis of the microarray data performed by Luckert et al. (2015) predicted an inhibition of the transcriptional regulators NR1I3 and HNF4A, whereas for NR0B2, an activation was predicted. In agreement with the results of the present study, Xiong et al. [39] observed a reduction in the gene expression of *Slc10a1* and *Slco1b2* in rat livers after treatment with senecionine (35 mg per kg body weight, 36 h). In contrast, no effect on the expression of *ABCB11* was observed in rat livers. However, after treatment with *Senecio vulgaris*, a PA-producing plant, Xiong et al. (2014) also observed a reduced gene expression of *Abcb11* and the SLC transporters *Slc10a1*, *Slco1a2*, and *Slco1b2* in rat livers [40]. A reduced expression of *Cyp7a1* and *Baat* was detected in rat livers after treatment with senecionine, whereas the expression of *Cyp8b1*, in contrast to the results presented here, was not affected [39]. After treatment with *Senecio vulgaris*, reduced gene expression of *Baat*, but not of *Cyp7a1* and *Cyp8b1*, was observed in rat livers [40]. Consistent with the results of the present study, the two in vivo studies mentioned above also described a reduced gene expression of *NR0B2* in rat livers after treatment with senecionine or *Senecio vulgaris* [39,40].

The reduced expression of the transporters *SLC10A1*, *SLC22A7*, *SLC22A9*, *SLC51A*, *SLCO1B1*, and *SLCO2B1*, determined in the present study, may possibly lead to a reduced uptake of bile acids, since these transporters are located in the basolateral membrane of hepatocytes and are responsible for the uptake of bile acids and other organic anions from sinusoidal blood. In contrast, the transporters ABCC2 and ABCB11 are located on the canalicular side, and are responsible for the secretion of bile acids into the bile ducts. Therefore, a reduced gene expression of these transporters may result in a reduced secretion of bile acids. In this context, in the present study, an impairment of hepatobiliary transport was detected in vitro via fluorescence microcopy. Similar effects were also observed in vivo in mice [17]. Furthermore, inhibition of ABCB11, in particular by accumulation of bile acids in the hepatocytes, has already been described for the development of cholestasis [25,41,42]. Regarding the PA-induced downregulation of various transcription factors, PAs may also affect bile acid, cholesterol homeostasis, and associated signaling pathways via interaction with transcriptional regulators. Furthermore, the observed PA-related downregulation of the enzymes *CYP27A1*, *CYP39A1*, *CYP7A1*, and *CYP8B1* may inhibit bile acid synthesis, which could be due to a possible feedback regulation induced by accumulating toxic bile acids. In addition, the decreased gene expression of *CYP3A4*, *BAAT*, *SULT2A1*, and *UGT2B4* indicates a potential decrease in bile acid metabolism and conjugation, which in turn would lead to a reduced detoxification of bile acids.

With regard to the downregulation of *CYP7A1*, the rate-limiting enzyme of bile acid synthesis, the present study used a reporter gene assay to investigate the inhibition of transcriptional activity of the *CYP7A1* promoter in the hepatocyte cell line HepG2. The observed inhibition of the *CYP7A1* promoter activity after PA treatment was not concentration-dependent. Additionally, the effect of PA treatment on the gene expression of *CYP7A1* in HepG2 cells was investigated. Whereas in the metabolically highly active HepaRG cells PA treatment caused a significantly reduced *CYP7A1* expression, in HepG2 cells, only a slight effect on the gene expression of *CYP7A1* was observed after 24 h of PA exposure. Since HepG2 cells generally exert low metabolic activity, the results suggest that the PA metabolites may be responsible for the reduced *CYP7A1* gene expression shown in HepaRG cells.

Impairment of tight junctions, changes in the cytoskeleton, oxidative stress, and induction of apoptosis/necrosis are also key events associated with the development of cholestasis [20,24,25,26]. The induction of apoptotic processes by PA treatment, both intrinsic and sensitized to extrinsic apoptosis, has been shown in various studies [29,43]. Damage to tight junctions can lead to an increase in paracellular permeability, reflux of bile constituents into the sinusoidal space of Disse and plasma, and a reduction of the osmotic gradient, which is normally the driving force for bile secretion [24,44,45]. With regard to a possible influence on the tight junction protein ZO-1, protein location was investigated by immunofluorescence staining. ZO-1 showed discontinuities in the staining only after 24 h of treatment with senecionine (35 µM), indicating a potential damage of the tight junctions. For the other three PAs echimidine, heliotrine, and senkirkine, no influence on ZO-1 could be detected, suggesting that this mechanism might only be marginally involved in a potential PA-associated cholestasis. To the best of our knowledge, no further studies investigating a PA-related effect on hepatocellular tight junction proteins are available.

A key event in the development of cholestasis is the accumulation of bile acids, leading to the induction of apoptosis/necrosis, oxidative stress, and a regulation of nuclear receptors and gene expression. Various in vivo studies have shown elevated bile acid levels in rat serum after administration of both senecionine and *Senecio vulgaris* [17,39,40]. The authors concluded that PA treatment resulted in an excess accumulation of bile acids in hepatocytes, which led to an adaptive response to avoid bile acid overload in hepatocytes. This adaptive response includes suppression of bile acid de novo synthesis (reduced gene expression of e.g., *Cyp7a1*), reduced sinusoidal reabsorption (reduced expression of e.g., *Slc10a1*), and reduction of bile acid accumulation by activation of alternative basolateral export pumps (Abcc3 and Abcc4). In the present in vitro study, a potential suppression of bile acid synthesis and sinusoidal uptake by regulation of corresponding enzymes and transporters (e.g., *SLC10A1* and *CYP7A1*) was shown in HepaRG cells at the mRNA level. However, no increased mRNA expression of alternative export pumps was detected, but, on the contrary, a downregulation of *ABCC3* was detected. In accordance with the studies of Xiong et al. (2014) [39,40], further studies showed an increase in bile acids in the serum and an increase in liver toxic enzyme markers and bilirubin in horses, as well as in calves, after ingestion of Senecio species or *Cynoglossum officinale* [46,47,48,49]. All of these data suggest that PA treatment leads to increased serum bile acid levels. However, in the present in vitro study, reduced bile acid levels were measured intra- and extracellularly in PA-treated cells compared to solvent-treated cells. The strength of the reduction is associated with the previously observed cytotoxicity [29] and the described potency of the tested PA [29,50]. A possible explanation for the observed reduced bile acid levels may be that PAs intracellularly disturb bile acid metabolism in HepaRG cells so dramatically that significantly less bile acids are formed and thus secreted.

In summary, we observed a strong PA-mediated impairment of bile acid homeostasis in human HepaRG cells, comprising a downregulation of numerous genes involved in bile acid uptake, synthesis, detoxification, and secretion. Furthermore, bile acid secretion seems to be disturbed, and thus might contribute to the development of cholestatic liver disease. Generally, the most pronounced effects were observed for senecionine and echimidine, representing PAs of the retronecine type. Using the HepaRG cell model and the molecular initiating and key events addressed in the AOP for cholestasis, we could confirm the evidence from in vivo studies for the development of cholestasis in our in vitro cell model.

## Figures and Tables

**Figure 1 foods-10-00161-f001:**
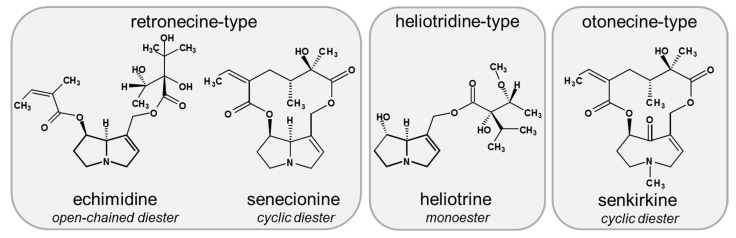
Chemical structure of PAs used in this study. Selected PAs represent the main occurring basic structures of necine base types and structures of esters: heliotrine is a monoester of the heliotridine-type, and senecionine and senkirkine are both cyclic diesters representing retronecine- and otonecine-type PAs, respectively. The open-chained diester echimidine belongs to the retronecine-type group.

**Figure 2 foods-10-00161-f002:**
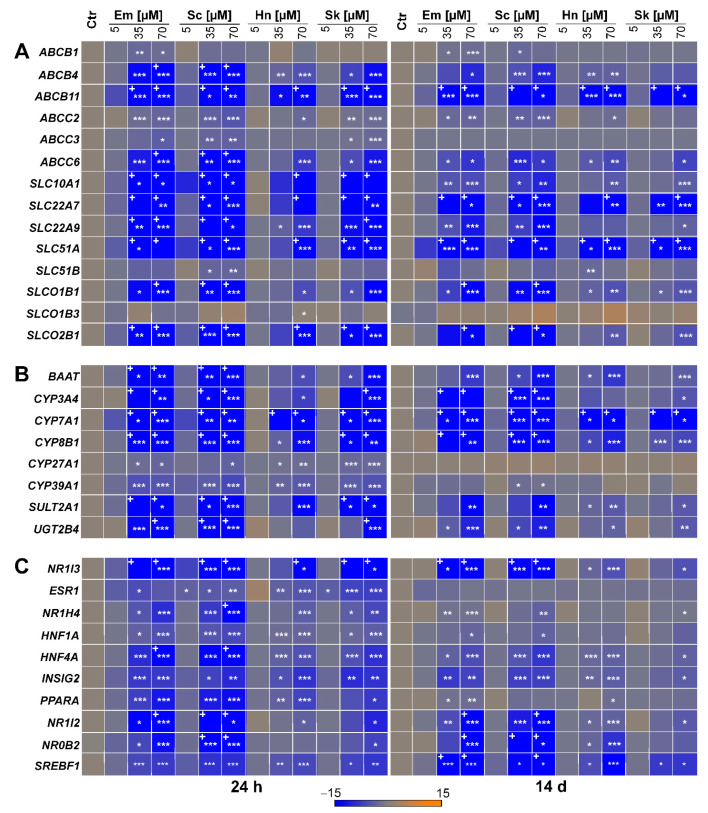
PA-dependent alterations of bile acid homeostasis-associated gene expressions ((**A**): transporters, (**B**): enzymes, (**C**): nuclear receptors) in HepaRG cells. HepaRG cells were seeded and cultivated as described in the material and methods section. Subsequently, the cells were incubated for 24 h or 14 days with 5, 35, or 70 µM of PA or the solvent (Ctr; 1.7% DMSO and 0.7% ACN). The total RNA was isolated and transcribed into cDNA. Expression analysis was performed by qRT-PCR. Gene expression of the target gene was normalized to *ACTB* and referred to solvent control to obtain relative expression (2^−ΔΔCt^ method). The heat map shows the relative expression values for up- and downregulation as positive and negative fold changes (means of three independent experiments with three replicates each). For better comparability, only the fold change range between −15 and 15 is shown. Values exceeding this range are marked with +. Statistical differences were evaluated using one-way ANOVA followed by Dunnett’s test: * *p* < 0.05, ** *p* < 0.005, *** *p* < 0.001. Em, echimidine; Sc, senecionine; Hn, heliotrine; Sk, senkirkine.

**Figure 3 foods-10-00161-f003:**
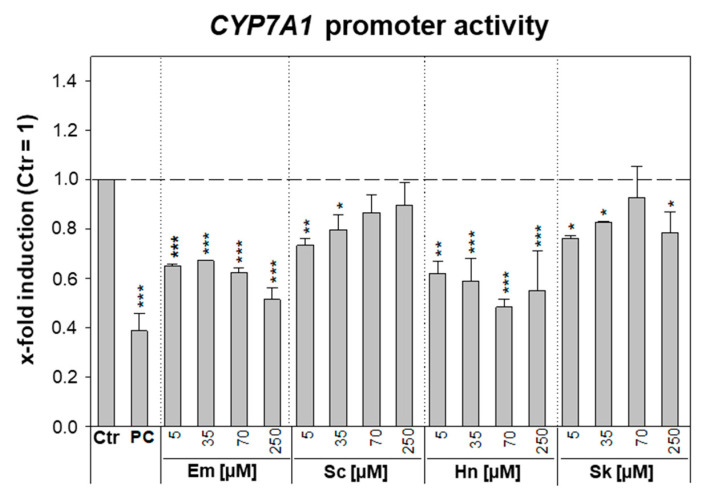
Interaction of PAs with *CYP7A1* promoter activity in HepG2 cells. HepG2 cells were cultivated and seeded as described in the materials and methods section. Cells were transfected with the reporter gene plasmid pGL4.14-CYP7A1-Prom (80 ng) and the control plasmid pcDNA3-Rluc (1 ng) for 6 h and subsequently treated with 5, 35, 70, or 250 µM of PA, solvent (Ctr; 2.5% ACN), or positive control (PC, 5 µM of PMA). After an incubation period of 24 h, the cells were lysed, and the firefly and Renilla luciferase activity was detected. The activity of the firefly luciferase was normalized to the activity of the Renilla luciferase and referred to solvent control (= 1) to obtain the x-fold induction. Shown are means ± standard deviations of three independent experiments with three replicates each. Statistical differences were evaluated using one-way ANOVA followed by Dunnett’s test: * *p* < 0.05, ** *p* < 0.005, *** *p* < 0.001. Em, echimidine; Sc, senecionine; Hn, heliotrine; Sk, senkirkine.

**Figure 4 foods-10-00161-f004:**
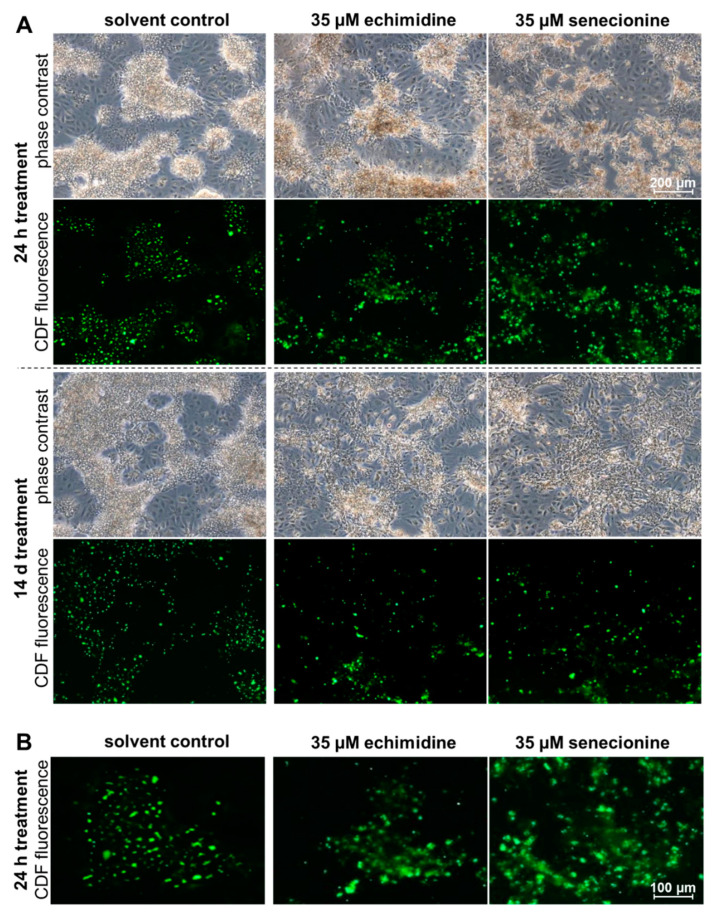
PA-dependent disturbance of ABCC2 driven efflux in HepaRG cells. (**A**) Differentiated HepaRG cells were incubated for 24 h or 14 days with 35 µM of echimidine, senecionine, or the solvent (1.7% DMSO and 0.35% ACN). To localize the bile canaliculi, the cells were incubated with 5 µM of 5(6)-carboxy-2′,7′-dichloro-fluorescein diacetate (CDFDA) for 30 min at 37 °C, and then analyzed on the fluorescence microscope Axio Observer.D1 (objective EC Plan-Neofluar 5x/0.16 Ph 1) under transmitted light and after excitation with 470 nm at 525 nm. The membrane-bound, non-fluorescent CDFDA is intracellularly converted by esterases into the green fluorescent ABCC2 substrate 5(6)-carboxy-2′,7′-dichlorofluorescein (CDF). By ABCC2-mediated transport, CDF enters the bile ducts. Representative sections are shown. The indicated scale applies to all images. (**B**) Exemplarily enlarged images of CDF fluorescence after treatment of HepaRG cells for 24 h with the solvent, 35 µM of echimidine, or 35 µM of senecionine. The indicated scale applies to all images of this enlargement.

**Figure 5 foods-10-00161-f005:**
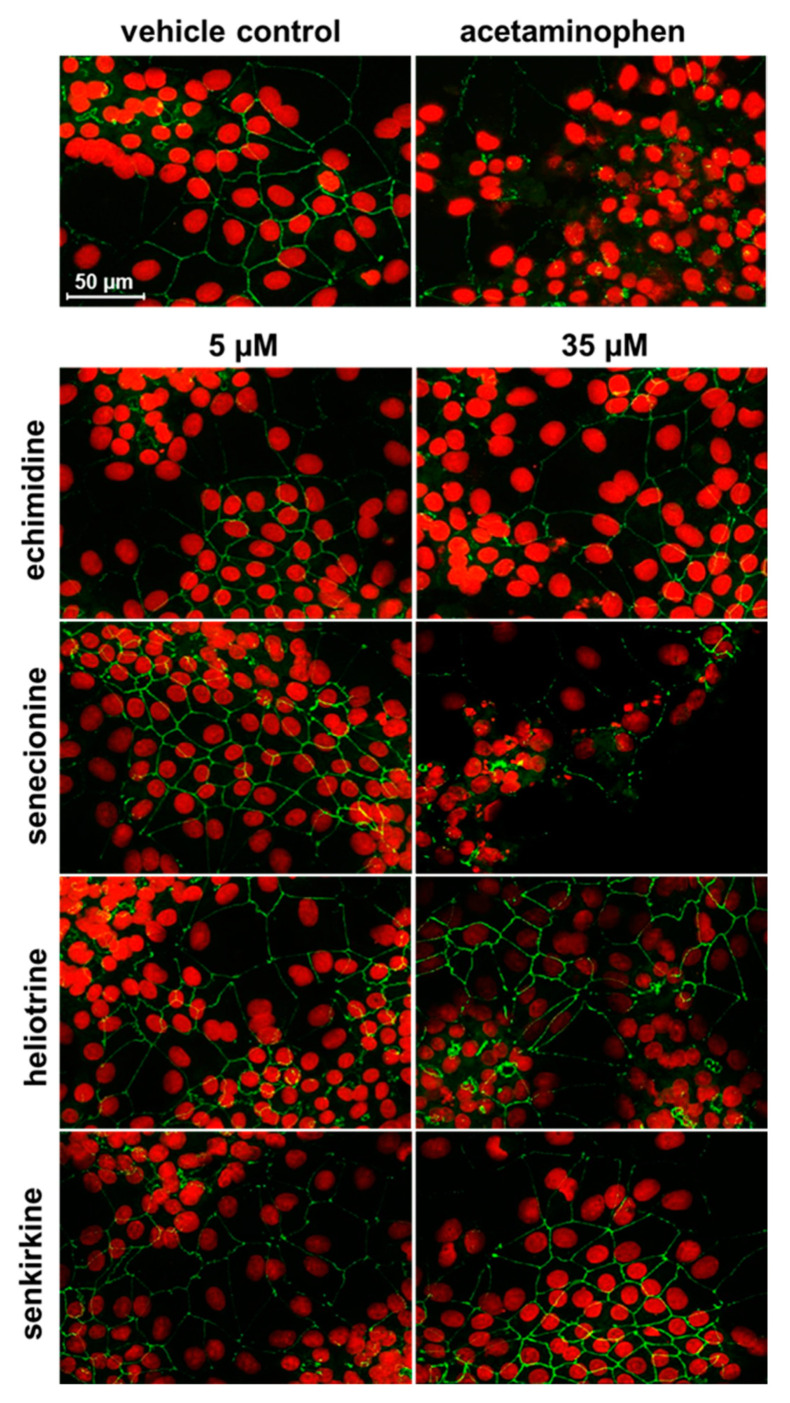
Effects of PA treatment on the tight junction protein ZO-1 in HepaRG cells. HepaRG cells were cultivated and seeded as described in the materials and methods section. After 14 days of further cultivation, the cells were treated with 5 or 35 µM of PA, solvent (1.7% DMSO and 0.35% ACN), or the positive control (10 mM of acetaminophen) for 24 h. Subsequently, the F-actin cytoskeleton was stained with ActinGreen 488 (green) and ZO-1 with an anti-ZO-1 antibody (1:500 in blocking solution) for 4 h. Afterwards, cells were treated with the secondary antibody Alexa Fluor 488 (anti-Rabbit IgG, 1:400 in blocking solution) for 1 h. Staining was determined using confocal fluorescence microscopy (Leica TCS SP5 with a 63× objective). Representative sections are shown. Indicated scale applies to all images.

**Figure 6 foods-10-00161-f006:**
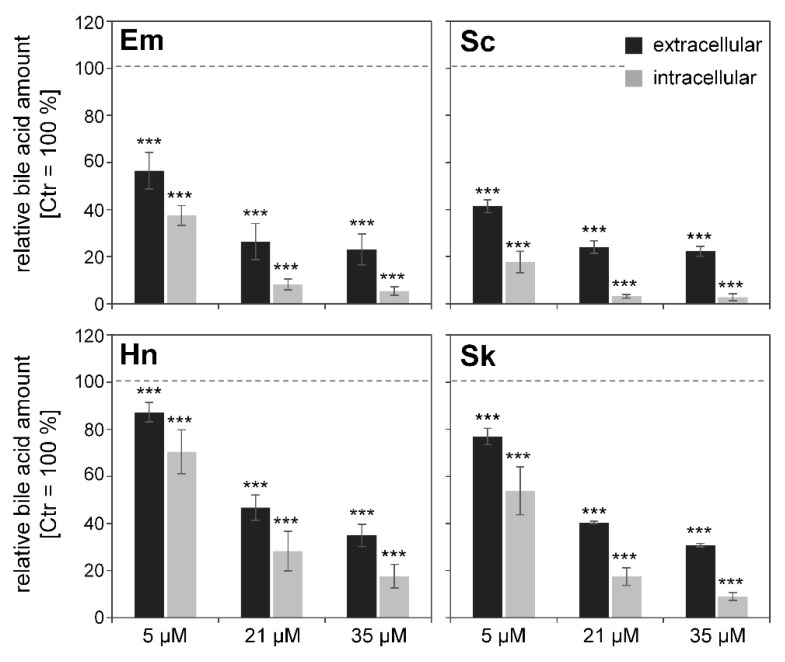
Sum of intra- and extracellular bile acid content after PA treatment. Differentiated HepaRG cells were treated under serum-free conditions for 48 h with PAs, as indicated in the figure. Bile acids were quantified as described in the material and methods section. The sum of bile acids was calculated and normalized to solvent control (1.7% DMSO, 0.35% ACN) set as 100%. As the control, cells were treated with 20 µM of known cholestasis inducer cyclosporine A, resulting in a decrease of bile acid amounts to 50% ± 15 in the medium and 20% ± 12 in the cells. Statistical differences were evaluated using one-way ANOVA followed by Dunnett’s test: *** *p* < 0.001.

## Data Availability

The data presented in this study are available in the Appendix A.

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
