# Peer review of "Pyrrolizidine Alkaloids Disturb Bile Acid Homeostasis in the Human Hepatoma Cell Line HepaRG"

_foods, 2021, doi:10.3390/foods10010161_

Round 1
Reviewer 1 Report
The manuscript concerned investigated the effect of selected pyrrolizidine alkaloids on bile acid homeostasis after exposure using the human hepatoma cell line HepaRG. Pyrrolizidine alkaloids are currently being studied extensively because of their toxic properties in animals. I believe that the manuscript describes interesting research from a food and feed safety point of view. The manuscript is well prepared.
Reviewer 2 Report
This manuscript discussed the effects of pyrrolizidine alkaloids on bile acid homeostasis from several perspectives, including alteration of gene expression, canalicular efflux, the impairment of tight junctions and bile acid content. The data is well presented and supportive. The clarity of the writing is exemplary. One question regarding the discussion.
The simultaneous decrease of extracellular and intracellular bile acids in HepaRG cells is consistent with the compromise of bile acid biosynthesis, which is also supported by the reduced CYP7A1 expression observed with qRT-PCR in HepaRG cells. The reduced CYP7A1 expression however is not as pronounced by dual luciferase reporter assay in HepG2 cells. The reviewer wondered whether authors could carry out the dual luciferase reporter assay in HepaRG cells to lend more support to the idea that PA metabolites are responsible for the reduced CYP7A1 expression.
A quick note, the legend in Figure 6 should be ‘intracellular’.
Reviewer 3 Report
The manuscript by Waizenegger et al., entitled “Pyrrolizidine alkaloids disturb bile acid homeostasis in the human hepatoma cell line HepaRG” investigated a pathway possibly involved in pyrrolizidine alkaloids (PA) toxicity causing disturbance of bile acid homeostasis. In this framework, authors evaluated the influence of four structurally different PA on bile acid homeostasis using the human hepatoma cell line HepaRG, which disturbed important biliary efflux mechanisms. This study may contribute to better understand the molecular mechanisms in the development of severe liver damage in PA intoxicated humans. Indeed, the impairment of bile acid secretion might contribute to the development of cholestatic liver disease. The article is well written and the experiment are well conducted.
-Page 1, line 17: change the abbreviation ‘d’ by ‘days’. Please check this in overall text;
-Page 3, line 107: change the abbreviation ‘ml’ by ‘mL’. Please check this in overall text;
-Page 3, line 110: the sentence is not clear, please rephrase.
